# Viral load Reduction in SHIV-Positive Nonhuman Primates via Long-Acting Subcutaneous Tenofovir Alafenamide Fumarate Release from a Nanofluidic Implant

**DOI:** 10.3390/pharmaceutics12100981

**Published:** 2020-10-17

**Authors:** Fernanda P. Pons-Faudoa, Nicola Di Trani, Antons Sizovs, Kathryn A. Shelton, Zoha Momin, Lane R. Bushman, Jiaqiong Xu, Dorothy E. Lewis, Sandra Demaria, Trevor Hawkins, James F. Rooney, Mark A. Marzinke, Jason T. Kimata, Peter L. Anderson, Pramod N. Nehete, Roberto C. Arduino, K. Jagannadha Sastry, Alessandro Grattoni

**Affiliations:** 1Department of Nanomedicine, Houston Methodist Research Institute, Houston, TX 77030, USA; fpons@houstonmethodist.org (F.P.P.-F.); nditrani@houstonmethodist.org (N.D.T.); antons.v.sizovs@gmail.com (A.S.); 2School of Medicine and Health Sciences, Tecnologico de Monterrey, Monterrey 64710, NL, Mexico; 3College of Materials Sciences and Opto-Electronic Technology, University of Chinese Academy of Science (UCAS), Shijingshan, Beijing 100049, China; 4Department of Comparative Medicine, Michael E. Keeling Center for Comparative Medicine and Research, MD Anderson Cancer Center, Bastrop, TX 78602, USA; kshelton1@mdanderson.org (K.A.S.); pnehete@mdanderson.org (P.N.N.); jsastry@mdanderson.org (K.J.S.); 5Department of Molecular Virology and Microbiology, Baylor College of Medicine, Houston, TX 77030, USA; zohakmomin@gmail.com (Z.M.); jkimata@bcm.edu (J.T.K.); 6Department of Pharmaceutical Sciences, Skaggs School of Pharmacy and Pharmaceutical Sciences, University of Colorado-Anschutz Medical Campus, Aurora, CO 80045, USA; lane.bushman@cuanschutz.edu (L.R.B.); peter.anderson@cuanschutz.edu (P.L.A.); 7Center for Outcomes Research and DeBakey Heart and Vascular Center, Houston Methodist Research Institute, Houston, TX 77030, USA; sxu@houstonmethodist.org; 8Weill Medical College of Cornell University, New York, NY 10065, USA; 9Academic Institute Houston Methodist, Houston, TX 77030, USA; dlewis3@houstonmethodist.org; 10Department of Radiation Oncology, Weill Cornell Medicine, New York, NY 10065, USA; szd3005@med.cornell.edu; 11Department of Pathology and Laboratory of Medicine, Weill Cornell Medicine, New York, NY 10065, USA; 12Gilead Sciences, Inc., Foster City, CA 94404, USA; trevor.hawkins@gilead.com (T.H.); jim.rooney@gilead.com (J.F.R.); 13Departments of Pathology and Medicine, Johns Hopkins University School of Medicine, Baltimore, MD 21224, USA; mmarzin1@jhmi.edu; 14The University of Texas MD Anderson Cancer Center UTH Health Graduate School of Biomedical Sciences, Houston, TX 77030, USA; 15Division of Infectious Diseases, Department of Internal Medicine, McGovern Medical School at The University of Texas Health Science Center at Houston, Houston, TX 77030, USA; roberto.c.arduino@uth.tmc.edu; 16Department of Thoracic Head and Neck Medical Oncology, University of Texas MD Anderson Cancer Center, Houston, TX 77030, USA; 17Department of Surgery, Houston Methodist Research Institute, Houston, TX 77030, USA; 18Department of Radiation Oncology, Houston Methodist Research Institute, Houston, TX 77030, USA

**Keywords:** HIV treatment, implantable drug delivery, viral load, TAF monotherapy, long-acting TAF

## Abstract

HIV-1 is a chronic disease managed by strictly adhering to daily antiretroviral therapy (ART). However, not all people living with HIV-1 have access to ART, and those with access may not adhere to treatment regimens increasing viral load and disease progression. Here, a subcutaneous nanofluidic implant was used as a long-acting (LA) drug delivery platform to address these issues. The device was loaded with tenofovir alafenamide (TAF) and implanted in treatment-naïve simian HIV (SHIV)-positive nonhuman primates (NHP) for a month. We monitored intracellular tenofovir-diphosphate (TFV-DP) concentration in the target cells, peripheral blood mononuclear cells (PBMC). The concentrations of TFV-DP were maintained at a median of 391.0 fmol/10^6^ cells (IQR, 243.0 to 509.0 fmol/10^6^ cells) for the duration of the study. Further, we achieved drug penetration into lymphatic tissues, known for persistent HIV-1 replication. Moreover, we observed a first-phase viral load decay of −1.14 ± 0.81 log_10_ copies/mL (95% CI, −0.30 to −2.23 log_10_ copies/mL), similar to −1.08 log_10_ copies/mL decay observed in humans. Thus, LA TAF delivered from our nanofluidic implant had similar effects as oral TAF dosing with a lower dose, with potential as a platform for LA ART.

## 1. Introduction

Advances in antiretroviral therapy (ART) have transitioned HIV-1 infection into a chronic disease. Daily ART regimens improve health, extend life and markedly reduce risk of HIV-1 transmission [1]. The 2020 90-90-90 treatment target by the UNAIDS is to diagnose 90% of all people living with HIV-1 (PLWH), out of which 90% will receive continual ART, and 90% of these will attain viral suppression. However, the treatment targets have not been met. In 2019 only 24.5 million out of 37.9 million PLWH had access to ART [2,3]. Further, ART adherence is an issue considering PLWH discontinue treatment due to pill burden, dosing frequency, food requirements and safety and tolerability concerns [4,5]. To address this issue, ART needs to shift to long-acting (LA) antiretrovirals to provide more convenient dosing, enhance tissue penetrance, improve resistance profiles and reduce toxicity [6].

Current LA ART undergoing clinical trials are once-weekly oral pills, and intramuscular and subcutaneous injectables. Islatravir, also known as MK-8591, is a nucleoside reverse transcriptase translocation inhibitor that suppresses plasma HIV-1 RNA by 1.2 log_10_ copies per mL for at least 7 days after a single 0.5 mg oral pill in treatment-naïve PLWH (NCT02217904) [7,8]. Nanoformulations of cabotegravir, an integrase strand transfer inhibitor, and rilpivirine, a non-nucleoside reverse transcriptase inhibitor, are undergoing safety and efficacy testing comparing 4 week and 8 week intramuscular injections in the LATTE-2 study (NCT02120352) [9,10]. Lenacapavir, formerly known as GS-6207, is an HIV capsid inhibitor undergoing safety and efficacy testing as a 6 month subcutaneous injection in combination with other ART in the CAPELLA and CALIBRATE trials (NCT04150068, NCT04143594) [11,12]. In contrast, most approved ART drugs have poor solubility and thus are not suitable for LA injectable formulations [13].

Implantable drug delivery systems are under development as another platform for LA ART [14]. A nonbiodegradable polymeric implant loaded with islatravir was subcutaneously administered to rats and uninfected nonhuman primates (NHP). The islatravir implant achieved clinically relevant drug concentrations in plasma and peripheral blood mononuclear cells (PBMC) for more than 6 months. However, islatravir implants have not undergone HIV treatment and prevention studies [15]. A cabotegravir formulation with 2-hydroxypropyl-β-cyclodextrin was loaded into a drug reservoir harboring a nanofluidic membrane and subcutaneously implanted in rats. This study evaluated pharmacokinetics and found drug levels were maintained above clinical values for 3 months [16]. Another notable drug candidate for LA platform is tenofovir alafenamide (TAF), a potent drug widely used as a first-line ART regimen in combination with other antiretrovirals. For HIV-1 pre-exposure prophylaxis (PrEP), TAF is under development as a single-agent LA in several drug delivery systems [17,18,19,20,21,22], however, none of these systems have yet been tested for HIV-1 treatment.

Here, we administered LA TAF monotherapy for the first time to simian HIV (SHIV)-infected NHP. The purpose of this study was to demonstrate that subcutaneous LA TAF delivered from our nanofluidic implant was similar to oral TAF dosing. In this 1-month study, we evaluated intracellular tenofovir diphosphate (TFV-DP) concentrations in PBMC and logarithmic viral load reduction after implantation in simian HIV (SHIV)-infected NHP. Further, we assessed TFV-DP concentrations in tissues of relevance to HIV-1 transmission and viral reservoirs. These data indicate the potential of our nanofluidic technology to transform HIV-1 therapeutic interventions as a versatile platform for LA ART.

## 2. Materials and Methods

### 2.1. Nanofluidic Membrane Fabrication and Characterization

Silicon nanofluidic membranes were fabricated using common semiconductor manufacturing processes. Detailed fabrication steps and parameters are available elsewhere [23,24]. Briefly, a silicon on insulator wafer was vertically etched via a deep reactive ion etching to obtain a tightly packed array of slit nanochannels in the device layer (10 µm). On the opposite side, the handle wafer (400 µm) was etched via deep reactive ion etching to create a 200 µm wide cylindrical microchannels. Both etches were stopped at the SiO_2_ insulator layer (1 µm) which was removed using a buffered oxide etchant solution. To achieve the desired nanochannel size, ~275 nm of silicon dioxide (SiO_2_) was generated using wet thermal oxidation. Then, a thin ~100 nm layer of silicon carbide (SiC) was placed on top, using plasma-enhanced chemical vapor deposition. Individual nanofluidic membranes (6 mm × 6 mm) were obtained dicing the original wafer with an ADT dicing saw (ADT 7100 Dicing Saw, Advanced Dicing Technologies, Zhengzhou, China). The complete nanofluidic membrane featured 278,600 monodisperse slit nanochannels organized in 199 groups of 1400 nanochannels. Each group was vertically connected to the opposite side microchannels, arranged in a hexagonal pattern for maximum structural stability.

The silicon nanofluidic membranes structure was assessed using scanning electron microscopy (SEM; Nova NanoSEM 230, FEI, Hillsboro, OR, USA) at the Microscopy—SEM core of the Houston Methodist Research Institute (HMRI), Houston, TX, USA. Nanochannel size was measured on membrane cross-sections obtained via gallium ion milling (FEI Dual-Beam 235 FIB, FEI, Hillsboro, OR, USA).

Nanochannel membranes were further characterized by filtering latex polystyrene 0.1, 0.2, 0.3 and 0.46 µm beads (Sigma Aldrich, St. Louis, MO, USA) through the membrane. We used a custom-made, two-part PEEK fixture that featured a sink and a drug reservoir [23]. The nanofluidic membrane was clamped between the two fixture parts with M3 screws and silicon gaskets to avoid leakage (McMaster Carr, Elmhurst, IL, USA). The sink reservoir was filled with 200 µL of 100 mM KCl with pH adjusted to 3 using 1 M HCl (*n* = 3). Stock particle solutions were made with 1% (*w/w*) beads in 100 mM KCl (pH 3) with 0.1% TritonX for particle stabilization. The drug reservoir was filled with 200 µL of the stock solution for each bead size. Particle concentration in the sink solution was measured after 5 days using a UV/Vis spectrophotometer (DU730, Beckman Coulter, Pasadena, CA, USA) at 252 nm.

SiO_2_ and SiC thicknesses and surface roughness (ρ) were determined by ellipsometry measurements using a J. A. Woollam M2000U ellipsometer (Lincoln, NE, USA).

### 2.2. Nanofluidic Implant Assembly

Oval-shaped medical-grade 6AI4V titanium implants measuring 20 × 13 × 4.5 mm (length × width × height) were machined using CNC milling (Groves Industrial Supply Corporation, Houston, TX, USA). The silicon membrane was glued to the implant using UV epoxy (OG116, Epoxy Technologies, Inc., Billerica, MA, USA) and cured for 2 h with a UV lamp (UVP UVL-18 EL Series, Analytik Jena US LLC, Upland, CA, USA). Implants were assembled and primed with 1X phosphate-buffered saline (PBS) as previously described [18] and were loaded with approximately 250 mg TAF fumarate. Implants were maintained in sterile 1X PBS in a hermetically sealed container until implantation shortly after preparation. The treatment nanofluidic implants loaded with TAF are referred to as nTAF_t_. TAF fumarate was provided in kind by Gilead Sciences, Inc. (Foster City, CA, USA).

### 2.3. Ethics Statement

All animal procedures were conducted at the AAALAC-I accredited Michale E. Keeling Center for Comparative Medicine and Research, the University of Texas MD Anderson Cancer Center (UTMDACC), Bastrop, TX, USA. All animal experiments were carried out according to the provisions of the Animal Welfare Act, PHS Animal Welfare Policy and the principles of the NIH Guide for the Care and Use of Laboratory Animals. All procedures were approved by the Institutional Animal and Care and Use Committee (IACUC) at UTMDACC (IACUC #00001749-RN00 approval date 19 September 2017—expiration date 19 September 2020). Indian rhesus macaques (*Macaca mulatta: n* = 6; 3 males and 3 females) of 2–4 years and 2–5 kg bred at this facility were used in the study. All animals had access to clean, fresh water always and a standard laboratory diet. Euthanasia of the macaques was performed using humane practices (IV pentobarbital) recommended by the American Veterinary Medical Association Guidelines on Euthanasia. Further, the senior medical veterinarian confirmed euthanasia by the absence of heartbeat and respiration.

### 2.4. Study Population

The macaques used in this study were enrolled in a SHIV prevention study as controls (*n* = 6). Once a week, these macaques were rectally challenged with SHIV_SF162P3_ for up to 10 weeks until initial detection of plasma SHIV RNA [18]. Macaques were weekly monitored for SHIV infection and infection was confirmed after a consecutive detection of plasma SHIV RNA. After 2 and 4 exposure challenges, 50 and 100% of macaques, respectively, tested SHIV-positive [18]. Upon confirmation of viral infection, animals were transferred to this treatment study. Macaques were implanted with nTAF_t_ a week after confirmed infection via a minimally invasive 1 cm dorsal skin incision on the left lateral side of the thoracic spine as previously described [18].

### 2.5. Blood Collection and Plasma and PBMC Sample Preparation

Blood collection and sample preparation were performed as previously described [17,18] to assess plasma viral load and intracellular TFV-DP concentrations in PBMC. Briefly, blood was drawn in EDTA-coated vacutainer tubes before implantation and then on the same timepoints that corresponded to the control cohort in the SHIV prevention study timeline [18]. Plasma was isolated from blood via centrifugation at 1200× *g* for 10 min at 4 °C and stored at −80 °C until analysis. Afterwards, PBMC were separated from the residual drug by standard Ficoll-Hypaque centrifugation with over 95% cell viability. Subsequently PBMC were counted and centrifuged at 400× *g* for 10 min at 4 °C. Next, PBMC pellet was lysed in 500 µL of cold methanol/water (70%/30%, *v/v*) and stored at −80 °C until analysis.

### 2.6. Pharmacokinetic Analysis of TFV-DP in PBMC

The pharmacokinetic (PK) profile of TFV-DP in PBMC was evaluated for a month after nTAF_t_ implantation. In total, 5 animals were evaluated for drug PK due to implant malfunction in 1 animal with reported TFV-DP concentrations below limit of quantitation. Quantification of intracellular TFV-DP concentrations in PBMC was executed as previously described using validated liquid chromatographic-tandem mass spectrometric (LC/MS-MS) analysis [25,26] at the Colorado Antiviral Pharmacology Lab at the University of Colorado Anshultz Medical Campus. The lower limit of quantitation (LLOQ) in the assay was 25 fmol/sample. Standards and quality control were included to 5 fmol/samples as previously described [26] in samples that required additional sensitivity.

### 2.7. Tissue TFV-DP Quantification

Tissues were collected at necropsy after a month of subcutaneous TAF treatment. To assess TFV-DP quantitation, 50- to 75-mg frozen tissue samples from urethra, cervix, rectum, tonsil, liver, spleen and lymphoid (axillary, mesenteric, inguinal and cervical lymph nodes) tissues were homogenized. PK analysis of tissue TFV-DP concentration was conducted by the Clinical Pharmacology Analytical Laboratory at the Johns Hopkins University School of Medicine via previously described LC-MS/MS analysis [18,26]. Briefly, isolated TFV-DP tissue was indirectly measured through enzymatic conversion to tenofovir (TFV) molecule [26]. Afterwards, TFV-DP concentrations were normalized by the weight of analyzed tissue [27]. The LLOQ in the assay was 5 fmol/sample. This method is validated for luminal tissue (rectal and cervical tissue) TFV-DP concentrations in compliance with the FDA, Guidance for Industry: Bioanalytical Method Validation recommendations [28]; the method is not validated for other tissues. Nonetheless, the remaining tissues were quantified using the same method.

### 2.8. Infection Monitoring by SHIV RNA in Plasma

Infection was monitored by detection of SHIV RNA in plasma using a modification [18] of previously described methods [29,30]. Infections were confirmed after a consecutive plasma viral load assay. All samples were tested in duplicate reactions and plasma viral loads were reported as viral RNA (vRNA) copies/mL of plasma. Standard curves were generated with 10-fold serial dilutions (1 to 1 × 10^6^ copies per reaction) of an in vitro transcribed SIV gag RNA. The assay was considered positive above the 50 copies/mL limit of detection.

### 2.9. TAF Stability Analysis in Drug Reservoir

TAF stability analysis in drug reservoir was done as previously described [18]. Briefly, filtered drug reservoir samples were analyzed via high-performance liquid chromatography (HPLC) on an Agilent Infinity 1260 system (Agilent Technologies Inc., Santa Clara, CA, USA) equipped with a diode array and evaporative light scattering detectors using a 3.5-µm 4.6 × 100 mm Eclipse Plus C18 column and water/methanol as the eluent and 25 µL injection volume. Peak areas were analyzed at 260 nm absorbance. Drug solids from within the implant were analyzed via UV-vis spectroscopy on a Beckman Coulter DU^®^ 730 system (Beckman Coulter, Pasadena, CA, USA). Peak areas were analyzed at 260 nm absorbance.

### 2.10. Assessment of Treatment nTAF Safety and Tolerability

Skin tissues were fixed in 10% buffered formalin and stored in 70% ethanol until analysis. Afterwards, tissues surrounding the implant site were embedded in paraffin, cut into 5 µm sections and stained with hematoxylin and eosin (H&E) at the Research Pathology Core HMRI, Houston, TX, USA. Semiquantitative histopathological assessment of inflammation surrounding the implant site was performed by 2 board-certified pathologists from 2 different institutions blinded to the group: Sandra Demaria, M.D. from Weill Cornell Medicine and Andreana L. Rivera, M.D. from Houston Methodist Hospital. The pathologists scored all slides using the inflammation scoring system Su et al. [31] adapted from the ISO published standard [32] (Appendix A). Briefly, samples are scored based on the presence of inflammatory cell and tissue characteristics. Afterwards, scores reported by both pathologists were averaged per implant (Appendix A) to calculate the total histological characteristic score for the nTAF_t_ group. Implant reactivity grade was calculated in accordance to Su et al. [22]. Next, the average placebo-adjusted implant reactivity score (S_pair_) was computed by subtracting the implant reactivity grade of nPBS [18] from nTAF_t_. nPBS was the control implant loaded with PBS used in these same macaques in the SHIV prevention study [18]. The S_pair_ classification of nTAF_t_ was determined per the published standard: minimal to no reaction (S_pair_ from 0.0 to 2.9), slight reaction (S_pair_ from 3.0 to 8.9), moderate reaction (S_pair_ from 9.0 to 15.0) and severe reaction (S_pair_ > 15.1) [22,31].

### 2.11. Statistical Analysis

All statistical analysis for calculation of the efficacy of TAF were performed with GraphPad Prism 8 (version 8.2.0; 2019, GraphPad Software, Inc., La Jolla, CA, USA) and STATA version 16 (StataCorp. 2019. Stata Statistical Software: Release 16. College Station, TX: StataCorp LLC). Data were represented as mean ± SD; median with interquartile range (IQR) between the first (25th percentile) and third (75th percentile) quartiles. Correlations were calculated with Spearman correlation coefficients. Outliers were detected with the Grubbs test (α = 0.05) and removed from descriptive statistics. Wilcoxon matched-pairs signed rank test was used to compare logarithmic viral load reductions to baseline. Generalized estimating equations (GEE) were used to examine the association of week or TFV-DP PBMC concentration with the change in logarithmic viral load reduction. An autoregressive correlation structure was specified for within-subject association between repeated measurements for subject. Statistical significance was defined as two-tailed *p* < 0.05.

## 3. Results

### 3.1. Nanofluidic Implant Assembly for Treatment

Our method to deliver long term acting ART leverages the unique properties of nanofluidic diffusion [32] to provide sustained drug release without the use of any active pumping mechanism [33,34]. The nanofluidic membrane 6 mm × 6 mm × 410 µm is epoxied within a medical-grade titanium drug reservoir (Figure 1A) and governs drug release from the implant [18,23]. Specifically, the concentration difference between the drug reservoir and the subcutaneous space drives drug diffusion across the membrane. The nanometric channel size and related interplay of steric and electrostatic forces acting on molecules diffusing through the nanochannels alters free Fickian diffusion [35]. As a result, a constant and sustained release that is quasi-independent of the concentration difference across the membrane [32,35] is achieved for a significant percentage of drug contained in the implanted reservoir (~85–95%). In this scenario, the release rate can be finely tuned by changing the nanochannels’ number and size [36]. Moreover, to maximize the volumetric loading efficiency, the drug reservoir was loaded with solid drug powder. As the interstitial fluid wets the membrane and enters the implant by capillary forces, the drug is solubilized and therefore able to diffuse through the membrane [37]. Thus, drug solubilization kinetics adds another layer of control over release rates from the implant.

In this study, we used nanofluidic membranes with an average nanochannel size of ~280 nm in accordance with the molecular size and physicochemical properties of TAF to achieve a suitable release rate [36]. Average nanochannel size was first assessed for every membrane by measuring the transmembrane nitrogen gas flow. In fact, transmembrane gas flow enables a nondestructive approach to membrane quality control [38]. Fitting the results with a previously developed algorithm [39], the average estimated size was 279 nm. We corroborated this result by measuring nanochannel dimensions from top-view (Figure 1B) images obtained via scanning electron microscopy (SEM) and cross-section (Figure 1C) obtained via FIB.

To further characterize the nanochannel size we measured the diffusion rate of monodisperse latex beads across the nanofluidic membrane (Figure 1D). The latex beads employed had an average diameter of 100, 200, 300 and 460 nm. Amounts of released particles were normalized to 100 nm beads. As expected, the beads release decreased with an increase of their respective size due to enhanced steric and hydrodynamic interaction with the nanochannels. Diffused amounts of 300 nm beads were significantly reduced (72% reduction) compared to 100 nm beads, while 460 nm beads were almost completely stopped.

Further, we assessed the thickness of the membrane layer and the surface roughness, before and after device implantation (Figure 1E,F). Ellipsometry measurement confirmed an average thickness of 275 nm for SiO_2_ and 100 nm for SiC. Both SiO_2_ and SiC showed no difference in thickness before and after implantation, confirming the inertness of the SiC outmost layer. SiC was purposely chosen as a membrane encapsulation layer because it offers biocompatibility and chemical inertness [40,41,42,43,44,45,46,47]. We previously demonstrated in vitro its ability to withstand a simulated physiological environment even at accelerated condition (77 °C) for up to 4 months [23]. Conversely, membrane surface roughness showed a significant increase after implantation, which can reasonably be associated with cellular debris or surface protein adsorption. Although protein adsorption on LA drug delivery systems is generally considered nonideal, we have extensively showed that the membrane function and drug release rates are not negatively affected by this process [18,48].

### 3.2. TFV-DP Concentration in PBMC and Tissues

For evaluation of TFV-DP concentration in PBMC and tissues, SHIV-positive rhesus macaques were subcutaneously implanted with nTAF_t_ (*n* = 5) in the dorsum for 1 month. Each macaque was implanted a week after confirmed seroconversion [18], thus time 0 denotes the day of implantation. At which, the median viral load prior to nTAF_t_ was 3.61 × 10^5^ copies/mL (95% CI, 3.11× 10^4^ to 9.47 × 10^6^). After implantation, TFV-DP PBMC concentrations were maintained at a median of 391.0 fmol/10^6^ cells (IQR, 243.0 to 509.0 fmol/10^6^ cells) for the duration of the study (Figure 2A). A maximum TFV-DP PBMC concentration was observed in the first week after implantation at a median of 408.0 fmol/10^6^ cells (IQR, 297.3 to 516.5 fmol/10^6^ cells); NHP 3 was determined an outlier by Grubbs. Thereafter, we observed median TFV-DP PBMC concentration of 480.0, 391.0 and 243.0 fmol/10^6^ cells (IQR, 269.0 to 634.5 fmol/10^6^ cells; IQR, 204.5 to 461.5 fmol/10^6^ cells; IQR, 179.0 to 481.0 fmol/10^6^ cells, respectively) within 2, 3 and 4 weeks after implantation, respectively.

We measured TFV-DP concentrations in tissues relevant to HIV-1 transmission or viral reservoirs after euthanasia (*n* = 5) (Figure 2B). Particularly, we assessed urethra, cervix, rectum, tonsil, liver, spleen, axillary lymph nodes (ALN), mesenteric lymph nodes (MLN), inguinal lymph nodes (ILN) and cervical lymph nodes (CLN). Drug penetration was observed at varying levels in all tissues after a month of subcutaneous TAF delivery (Figure 2B). Urethral, cervical and rectal tissues had lower median TFV-DP concentrations of 21.65 fmol/mg (IQR, 18.15 to 25.15 fmol/mg), 20.02 fmol/mg (IQR, 10.08 to 50.36 fmol/mg) and 46.64 fmol/mg (IQR, 22.14 to 59.34 fmol/mg), respectively. Tonsil, liver and spleen had higher median TFV-DP concentrations of 132.6 fmol/mg (IQR, 109.1 to 225.6 fmol/mg), 95.99 fmol/mg (IQR, 40.2 to 160.2 fmol/mg) and 225.0 fmol/mg (IQR, 132.6 to 247.7), respectively. Notably, the highest median TFV-DP concentrations were observed in the lymph nodes fmol/mg, with 377.0 fmol/mg (IQR, 156.6 to 1035 fmol/mg) in ALN, 271.7 fmol/mg (IQR, 134.0 to 417.5 fmol/mg) in MLN, 222.3 fmol/mg (IQR, 141.6 to 363.1 fmol/mg) in ILN and 306.1 fmol/mg (IQR, 206.6 to 491.0 fmol/mg) in CLN. Specifically, high lymphatic tissue concentrations are important since lymphatic viral persistence has been observed in humans with undetectable viral loads in blood [49].

### 3.3. nTAF Treatment Viral Load Reduction

To evaluate treatment efficacy of LA TAF, we assessed the logarithmic viral load reduction in SHIV-infected NHP implanted with nTAF_t_ (*n* = 6) (Figure 3A). After continuous TAF subcutaneous delivery, the first-phase change in plasma SHIV RNA compared to baseline levels was assessed individually for each NHP. NHP 1 had a first-phase SHIV RNA decay slope of −0.55 log_10_ copies/mL 4 days post-implantation. NHP 2 and 3 had a first-phase SHIV RNA decay slope of −0.30 and −1.71 log_10_ copies/mL, respectively, 5 days post-implantation. NHP 4 and 5 had a first-phase SHIV RNA decay slope of −0.90 and −2.23 log_10_ copies/mL a week post-implantation. Interestingly, NHP 6 had TFV-DP concentrations below LLOQ throughout the study yet had a first-phase SHIV RNA decay slope of −1.84 log_10_ copies/mL 5 days post-implantation. Overall, the mean first-phase change in SHIV RNA was −1.14 ± 0.81 log_10_ copies/mL (95% CI, −0.30 to −2.23 log_10_ copies/mL; *p* = 0.031).

Throughout 2 weeks of sustained TAF delivery, the mean viral load reduction compared to baseline levels was −1.19 ± 0.50 log_10_ copies/mL (95% CI, 0.19 to −2.57 log_10_ copies/mL; *p* = 0.063). However, the mean change in SHIV RNA from baseline reduced in 3 weeks to −0.67 ± 0.54 log_10_ copies/mL (95% CI, 0.004 to −1.34 log_10_ copies/mL; *p* = 0.063). Subsequently, the mean change in SHIV RNA increased logarithmic viral load reduction to −0.76 ± 0.70 log_10_ copies/mL (95% CI, 0.11 to −1.63 log_10_ copies/mL; *p* = 0.063) until study endpoint. Further, in Figure 3A there was a quadratic relationship between the week and the change in logarithmic viral load reduction (*p* < 0.001). The estimated change that fitted the trend line was −0.86, −1.26, −1.24 and −0.80 for week 1, 2, 3 and 4, respectively.

In addition, an intraindividual negative correlation coefficient between TFV-DP PBMC concentration and plasma viral load was observed in all NHP (Figure 3B–F). Viral load decreased per increasing intracellular TFV-DP concentration in PBMC. However, none had a Spearman correlation coefficient that was statistically significant. Moreover, an interindividual analysis demonstrated an inverse relationship between TFV-DP PBMC concentration and initial plasma viral load. For every one increase in TFV-DP PBMC concentration, the log_10_ (plasma viral load) would decrease by 0.0011 (95% CI: −0.0019, −0.0003, *p* = 0.009). Thus, a greater viral load reduction was observed in NHP with lower initial plasma viral loads.

### 3.4. TAF Stability in Drug Reservoir

To evaluate drug stability in nTAF_t_ after 1 month of in vivo implantation, we extracted residual contents from the implant and analyzed for TAF and its hydrolysis products (TAF*) (Table 1). Residual drug within the implant ranged 45.31–76.34% of the initial loaded amount. Implant nTAF_t_ from NHP 6 was removed from descriptive residual statistics because it malfunctioned and released 2.13% of the initial loaded amount. Further, TAF* within the implant was composed of TAF hydrolysis products, including TFV, with TAF stability ranging 41.29–61.86%. In contrast, a 96.71% TAF stability was observed in nTAF_t_ from NHP 6. Thus, the increased stability was likely attributed to negligible TAF release [37]. The nTAF_t_ implants had a mean release rate of 3.07 ± 1.10 mg/day, which was enough to create a 1.14 ± 0.81 viral load log reduction within the first week.

### 3.5. Histological Assessment of nTAF_t_ Safety and Tolerability

To assess nTAF_t_ safety and tolerability, we examined the tissue surrounding the implants via histopathological analysis (Figure 4). Specifically, to evaluate tissue response to the implant and the drug, we examined the fibrotic capsule in contact with the titanium reservoir or TAF-releasing nanofluidic membrane. In the most ideal scenario for this analysis, implants and tissues would have been fixed and sectioned together [31]. However, analysis of drug residual within the implant was crucial for assessing stability and in vivo release rate. Further, by considering the challenges associated with cutting through titanium reservoir and silicon membrane, this was not feasible, and implant required removal prior to tissue histological processing. Nevertheless, it is important to note that during harvesting, after performing an incision in the fibrotic tissue, implants freely slipped out of the capsule. No cell adhesion was observed, indicating that the extraction of implants did not damage or remove cells and tissues at the interface between implant and fibrotic capsule.

Histopathological scoring of five implants via hematoxylin and eosin (H&E) reported a total histological characteristic score of 16.1 ± 2.0 (scale from 0 to 31) and an average implant reactivity score of 27.1 (scale 0 to 56) (Appendix A). The calculated S_pair_ after a month of subcutaneous TAF delivery was 14.1, indicating a moderate reaction. This reaction could be attributable to the TAF release rate of 3.07 ± 1.10 mg/day. Interestingly, Su et al.’s [22] Generation B TAF implants reported severe reactions with 20 times less dose than nTAF_t_. The histopathological assessment demonstrated moderate inflammation in the surrounding subcutaneous tissue in contact with the titanium reservoir (Figure 4A) and the underlying skeletal muscle near the TAF-releasing nanofluidic membrane (Figure 4B). Further, as expected in foreign body reaction to implants, macrophages composed the majority of inflammatory cells [50]. To exclude macrophage recruitment from infection, the fibrotic capsule in contact with the reservoir and drug-eluting side were analyzed for fungi and bacteria via Grocott methenamine silver staining (Figure 4C,D) and acid-fast bacteria (AFB) (Figure 4E,F), respectively, and resulted negative.

## 4. Discussions

Our work represents the first report of treatment efficacy of an implantable LA ARV platform. In this 1-month monotherapy study in treatment-naïve SHIV-positive macaques, subcutaneous administration of TAF achieved high intracellular TFV-DP PBMC concentrations. Further, the nTAF_t_ implants had a mean release rate of 3.07 ± 1.10 mg/day, which demonstrated viral load reduction in treatment-naïve macaques with SHIV infection. After a month of implantation, the nTAF_t_ had a moderate reaction compared to the slight reaction observed in NHP implanted with nTAF for 4 months [18]. This was expected since treatment regimens require higher doses than prevention regimens.

Given that the NHP received a significantly lower dose of TAF, lower intracellular TFV-DP PBMC concentrations were detected than the 8.2 µM recorded in subjects after 14 days of 40 mg/day dose [6]. After 14 days of continuous TAF delivery at a release rate of approximately 3 mg/day, TFV-DP PBMC concentrations achieved a median of 0.96 nM (IQR, 0.54 to 1.27 nM). However, with 3.07 ± 1.10 mg/day of TAF, a smaller dose than in Descovy^®^ (200 mg emtricitabine/25 mg TAF), we showed the mean first-phase change in SHIV RNA was −1.14 ± 0.81 log_10_ copies/mL (95% CI, −0.30 to −2.23 log_10_ copies/mL). We exhibited similar viral reduction (−0.94 and −1.08 log_10_ copies/mL) with 2- and 5-times less TAF than PLWH treatment-naïve subjects receiving 8 mg/day and 25 mg/day, respectively [51]. Moreover, the observed −1.14 viral load reductions within a week of continuous TAF exposure offers an early measure of long-term responses. Thus, changes in viral load levels as early as 6 days after treatment initiation can correlate with virologic response in the long-term [52]. Furthermore, in a clinical scenario, HIV-1 RNA levels and the rate at which changes in viral load emerges can be insightful into individual ART relative efficacy [6,53]. Likewise, our short-term subcutaneous TAF delivery study viral reduction correlates with results from the 10-day oral TAF dosing which reported no resistance mutations known to reduce susceptibility to TFV [6,51].

Another key point in this study is that lymphatic tissues achieved the highest TFV-DP concentration after a month of sustained subcutaneous TAF delivery from our nanofluidic device. Our device released TAF at 3.07 ± 1.10 mg/day, a dose that was enough to show a viral load reduction in systemic blood and penetrate lymphatic tissues. Thus, by delivering TAF into the subcutaneous space, TAF bioavailability increases as it bypasses oral absorption kinetics and first-pass metabolism [54]. Further, with direct access to blood and circulatory systems, the nanofluidic membrane permits constant subcutaneous TAF release, increasing drug exposure to PBMC and lymphatic tissues [54,55]. Therefore, our nanofluidic platform addresses the concern of low intracellular drug concentrations in lymph nodes and lymphatic tissues with oral ART. Consequently, offering a solution to viral rebounds from low ART lymphatic tissue penetration in PLWH that discontinue ART [49]. Specifically, our nTAF_t_ implant would continuously expose the lymphatic system to TAF, thus possibly overcoming viral persistence in lymphatic tissues.

The present study was limited by the number of animals. However, the variation in first-phase viral load decay between animals warrants further investigations on optimizing the treatment for individuals with different initial viral loads. Although restricted by scheduled NHP blood collections to another study [18], similar timespans would offer additional statistical analysis. In addition, the study was limited with only one timepoint of tissue TFV-DP levels. Another earlier timepoint would offer insight on comparing tissue TFV-DP penetration after nTAF_t_ implantation. Notably, in this study we assessed the ability of sustained TAF monotherapy to reduce the viral load. However, it is well understood that multi-drug regimens are required for HIV treatment for efficacy and to mitigate the risk of the onset of drug resistance [6,51]. Therefore, for clinical relevance, in our future efforts we will focus on the implementation of the technology in the context of a multi-drug therapy.

## 5. Conclusions

In summary, our study provides evidence for the efficacy of the nTAF_t_ as a potential LA ART platform for sustained subcutaneous TAF delivery. The approach shows promise in obviating patient adherence issues. In harnessing nanotechnology to continuously deliver TAF at a lower dose, our implant succeeded in reducing the viral load similarly to daily oral TAF regimens. Our results suggest that by sustained constant subcutaneous administration, a lower dose for TAF could be used. In principle this could reduce cost of treatment and adverse side effects observed with higher doses of TAF. Furthermore, our nanofluidic implant demonstrated the proof-of-principle that sustained subcutaneous delivery of ART enhances lymphatic tissue penetration. Clinical deployment of this nanofluidic approach for HIV treatment would require a multi-drug regimen. Thus, further studies are warranted. However, our nanotechnology platform shows promise for use in HIV-1 LA ART and improve therapeutic outcomes in PLWH.

## Figures and Tables

**Figure 1 pharmaceutics-12-00981-f001:**
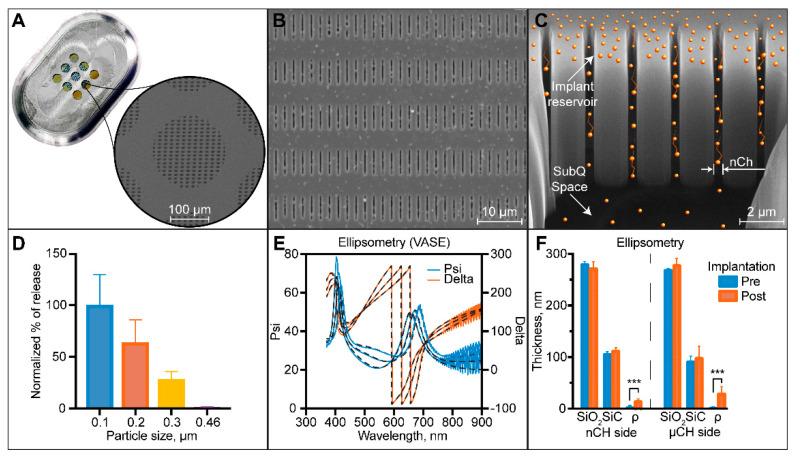
The nanofluidic implant for subcutaneous tenofovir alafenamide (TAF) HIV-1 treatment delivery (nTAF_t_). (**A**) Top-view image of the assembled nTAF_t_ with a zoom-in on the nanofluidic membrane demonstrating an SEM of the nanochannels. (**B**) Higher magnification of a top-view of SEM image of nanochannel membrane. (**C**) FIB image of nanochannel membrane cross-section showing drug release through vertically etched nanochannels. (**D**) Latex bead filtration through nanochannel. (**E**) Ellipsometry analysis of membrane surface. (**F**) Ellipsometry analysis of membrane surface.

**Figure 2 pharmaceutics-12-00981-f002:**
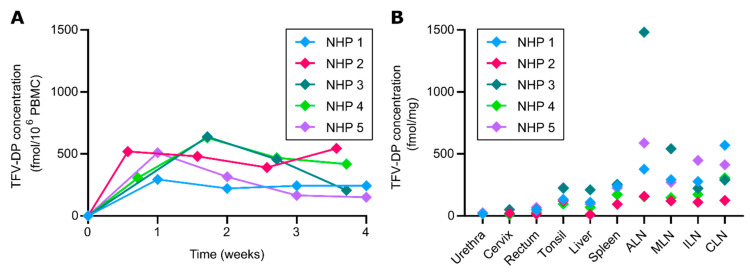
Tenofovir-diphosphate (TFV-DP) peripheral blood mononuclear cells (PBMC) and tissue distribution of TAF from subcutaneous nTAF_t_. nTAF_t_ implants (*n* = 5) were retrieved after 1 month. (**A**) Intracellular TFV-DP PBMC concentrations of nTAF_t_. (**B**) Tissue TFV-DP concentrations upon euthanasia after 1 month of implantation. NHP, nonhuman primate, ALN, axillary lymph nodes, MLN, mesenteric lymph nodes, ILN, inguinal lymph nodes, CLN, cervical lymph nodes.

**Figure 3 pharmaceutics-12-00981-f003:**
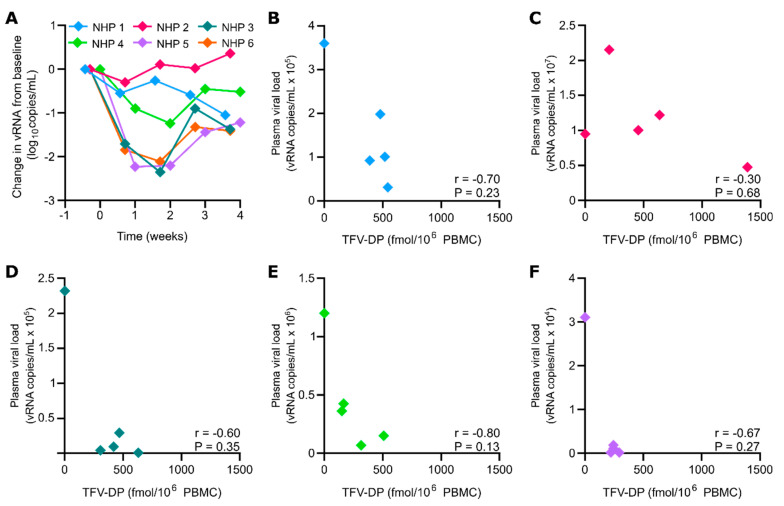
Viral load reduction and correlation with TFV-DP PBMC concentration. (**A**) Viral load reduction from baseline of nTAF_t_ (*n* = 6). Change in SHIV RNA from baseline (log_10_ copies/mL) at week 1, 2, 3 and 4 throughout continuous subcutaneous TAF dosing. Data are presented for each individual NHP. (**B**) NHP 1 TFV-DP PBMC concentration correlation with plasma viral load. (**C**) NHP 2 TFV-DP PBMC concentration correlation with plasma viral load. (**D**) NHP 3 TFV-DP PBMC concentration correlation with plasma viral load. (**E**) NHP 4 TFV-DP PBMC concentration correlation with plasma viral load. (**F**) NHP 5 TFV-DP PBMC concentration correlation with plasma viral load. Statistical analysis on panels (**B**–**F**) performed by Spearman correlation.

**Figure 4 pharmaceutics-12-00981-f004:**
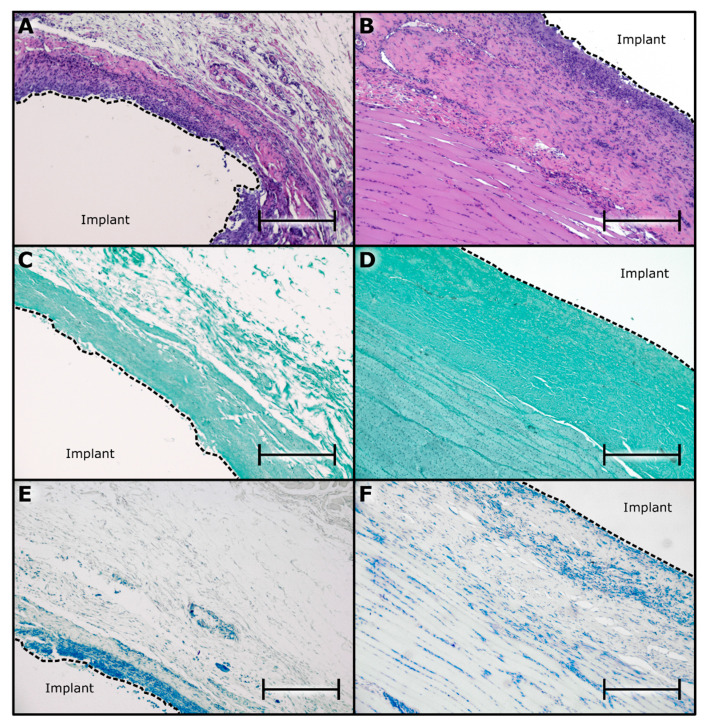
Histological inflammatory response to nTAF_t_. (**A**) Representative H&E stain of NHP skin surrounding nTAF_t_ in contact with the titanium reservoir. (**B**) Representative H&E stain of NHP skin surrounding nTAF_t_ near the TAF-releasing nanofluidic membrane. (**C)** Grocott methenamine silver stain of NHP skin surrounding nTAF_t_ in contact with the titanium reservoir. (**D**) Grocott methenamine silver stain of NHP skin surrounding nTAF_t_ near the TAF-releasing nanofluidic membrane. (**E**) Acid-fast bacteria stain of NHP skin surrounding nTAF_t_ in contact with the titanium reservoir. (**F**) Acid-fast bacteria stain of NHP skin surrounding nTAF_t_ near the TAF-releasing nanofluidic membrane. All images taken at 10 × magnification with 400 µm scale bar.

**Table 1 pharmaceutics-12-00981-t001:** nTAF_t_ drug residual after 1 month implantation.

nTAF_t_ (NHP #)	TAF Loaded (mg)	Residual TAF * (mg)	TAF Stability (%)	TAF Release Rate (mg/day)
1	244.1	173.36	61.86	2.53
2	257.4	191.64	41.29	2.35
3	244.1	110.61	47.50	4.77
4	248.5	149.06	55.42	3.55
5	253.0	193.15	48.27	2.14
6	276.3	270.41	96.71	0.21

* Hydrolysis products.

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
