# Peer review of "Viral load Reduction in SHIV-Positive Nonhuman Primates via Long-Acting Subcutaneous Tenofovir Alafenamide Fumarate Release from a Nanofluidic Implant"

_pharmaceutics, 2020, doi:10.3390/pharmaceutics12100981_

Round 1

Reviewer 1 Report

An excellent paper. Very well designed, executed and presented. The Introduction describes the background and purpose of the study. Methods are thorough. Results and discussion are excellent. Figures are of a high standard. Conclusions are fitting. 

Spelling error on Page 6 Line 247 - should be created?

Page 7 Line 291 should be PBMC

One question that I have is that from my experience is that subcutaneous implants are best suited to drugs that only require mcg per day. Eg. contraceptive hormones. Therefore, you can leave Implanon in the body for very long times without intervention. Your implant is at +2mg/day. So what is the maximum life span of your device? Is a short product lifespan a concern?

Author Response

This manuscript reports the effectiveness of the tenofovir alafenamide fumarate (TAF) treatment in the form of long term acting nanofluidic implant against SHIV infection in Indian rhesus macaques. The results are novel and interesting, and the text is well written. I have some minor concerns as described below.

Comment 1: 1. In 2.4. Study population (page 4 lines 154-162), it might be better to describe the viral dose per intrarectal challenge. I suppose SHIV-SF162P3 is a CCR5-tropic strain that causes chronic AIDS if macaques are untreated. Is that right?

Response 1: We appreciate the reviewer’s comment, however we did not include this information in the Materials and Methods to avoid confusion with the readers. In the study described in this manuscript no macaques were rectally challenged. These animals were previously challenged when they were controls in another study and we refer to that manuscript for viral dosage per challenge. The Reviewer is correct in their assumption, SHIV-SF162P3 is a CCR5-specific chimeric SHIV virus. The disease course after infection is similar to HIV-1 infection if untreated.

Comment 2: 2. In 3.1 Nanofluidic implant assembly for treatment (page 6):

“…the release rate can be finely tuned changing nanochannels number and size” (lines 251-252). Does this read “… by changing…”? “…diffuse trough the membrane” -> “diffuse through the membrane” (line 255).

Response 2: We thank the reviewer for noticing the spelling errors. These have been corrected.

Comment 3: The data on tissue distribution is interesting (page 8, lines 310-324). I wonder whether there are some previously published data on the tissue distribution of TFV-DP that can be compared to the present data. I suppose such discussion could result in a helpful discussion.

Response 3: We appreciate the reviewer’s comment and agree that tissue distribution of TFV-DP from a subcutaneous implant is interesting and it would greatly benefit from a comparison. However, there is no previously published data on tissue distribution of TFV-DP in nonhuman primates administered subcutaneous TAF that can be compared to the present data. Massud et al. (Journal of Infectious Diseases 2019) examined oral TAF in NHP but only assessed TFV-DP concentrations at 24 hours in vaginal and rectal tissue. Further, this data cannot be compared because NHP received oral TAF in conjunction with another antiretroviral, emtricitabine.

Comment 4: In 3.3 nTAF treatment viral load reduction, it seems that the treatment was effective in most of the animals tested. I have some suggestions for the section and Figure 3. All the viral load data from week 0 until week 4 would be appreciated, though these are shown in Fig 3B-F without the time point information.

Response 4: We appreciate the reviewer’s suggestions for the section 3.3 and Figure 3. The first-phase viral load decay demonstrates the change in

Comment 5: The term “first-phase viral load decay” in the study setting might need to be defined somewhere. Note that this is also used in Abstract.

Response 5: We thank the reviewer for the comment, however “first-phase viral load decay” is explained in section 3.3. “After continuous TAF subcutaneous delivery, the first-phase change in plasma SHIV RNA compared to baseline levels was assessed individually for each NHP.” (Page 8, lines322-324).

Comment 6: The correlation analyses are limited to intra-individual comparisons, but in my opinion there appears to be an inverse correlation between the initial viral load and the magnitude of log10 vRNA reduction if an inter-individual analysis is done (Fig. 3B-F). So, the different treatment efficacies observed in six macaques could be mostly due to the different initial viral loads they had. In Discussion, the authors list the variable viral loads as a limitation of the study (page 11, lines 436-437). However, as described above, this could be seen as a merit, as this suggests the need for further investigations on optimizing the treatment for individuals having different viral loads. I hope the authors could report the correlation analysis results.

Response 6: We appreciate the Reviewer’s comment and performed an inter-individual correlation analysis between TFV-DP PBMC concentration and initial plasma viral load. The following sentence was added “Moreover, an inter-individual analysis demonstrated an inverse relationship between TFV-DP PBMC concentration and initial plasma viral load. For every one increase in TFV-DP PBMC concentration, the log10 (plasma viral load) will decrease by 0.0011 (95% CI: -0.0019, -0.0003, p=0.009).  Thus, a greater viral load reduction was observed in NHP with lower initial plasma viral loads.” (Page 8, lines 343-347). Consequently, we agree with the Reviewer that the different treatment efficacies observed in six macaques could be mostly due to the different initial viral loads they had. The Discussion now reads “The present study was limited by the number of animals. However, the variation in first-phase viral load decay between animals warrants further investigations on optimizing the treatment for individuals with different initial viral loads.” (pages 11-12, lines 435-437).

Comment 7: As such, I am not sure of the way of correlation analyses between log10 vRNA and lin TFV-FP, as a log10 increase in vRNA actually means a 10-fold increase in the viral burden that may not be rescued by a 2-3 fold increase in TFV-DP distribution. I suppose the presentation of the data on the relationship between pharmacokinetics and viral kinetics could be somewhat improved.

Response 4: We agree with the Reviewer’s suggestion and changed Fig. 3B-F to show the correlation analysis between linear vRNA and linear TFV-DP. After reanalyzing the correlations, there was no change in the statistical analysis results.

Comment 8: The reference No. 18 seems to be the authors’ preprint published in bioRχiv. The preprint server name could be given.

Response 8: We thank the Reviewer for noticing this, the preprint server name is now given.

Comment 9: The reference No. 52 might need a correction.

Response 9: We thank the Reviewer for noticing this, the reference has been corrected.

Comment 10: Some words may be missing in the last sentence of Conclusions (“… shows promise for use in in HIV-1 LA…”) (page 12, lines 457-458).

Response 10: We thank the Reviewer for noticing the error, it has been corrected.

Reviewer 2 Report

This manuscript reports the effectiveness of the tenofovir alafenamide fumarate (TAF) treatment in the form of long term acting nanofluidic implant against SHIV infection in Indian rhesus macaques. The results are novel and interesting, and the text is well written. I have some minor concerns as described below.

  1. In 2.4. Study population (page 4 lines 154-162), it might be better to describe the viral dose per intrarectal challenge. I suppose SHIV-SF162P3 is a CCR5-tropic strain that causes chronic AIDS if macaques are untreated. Is that right?
  2. In 3.1 Nanofluidic implant assembly for treatment (page 6):
    • “…the release rate can be finely tuned changing nanochannels number and size” (lines 251-252). Does this read “… by changing…”?
    • “…diffuse trough the membrane” -> “diffuse through the membrane” (line 255).
  3. The data on tissue distribution is interesting (page 8, lines 310-324). I wonder whether there are some previously published data on the tissue distribution of TFV-DP that can be compared to the present data. I suppose such discussion could result in a helpful discussion.
  4. In 3.3 nTAF treatment viral load reduction, it seems that the treatment was effective in most of the animals tested. I have some suggestions for the section and Figure 3.
    • All the viral load data from week 0 until week 4 would be appreciated, though these are shown in Fig 3B-F without the time point information.
    • The term “first-phase viral load decay” in the study setting might need to be defined somewhere. Note that this is also used in Abstract.
    • The correlation analyses are limited to intra-individual comparisons, but in my opinion there appears to be an inverse correlation between the initial viral load and the magnitude of log10 vRNA reduction if an inter-individual analysis is done (Fig. 3B-F). So, the different treatment efficacies observed in six macaques could be mostly due to the different initial viral loads they had. In Discussion, the authors list the variable viral loads as a limitation of the study (page 11, lines 436-437). However, as described above, this could be seen as a merit, as this suggests the need for further investigations on optimizing the treatment for individuals having different viral loads. I hope the authors could report the correlation analysis results.
    • As such, I am not sure of the way of correlation analyses between log10 vRNA and lin TFV-FP, as a log10 increase in vRNA actually means a 10-fold increase in the viral burden that may not be rescued by a 2-3 fold increase in TFV-DP distribution. I suppose the presentation of the data on the relationship between pharmacokinetics and viral kinetics could be somewhat improved.
  5. The reference No. 18 seems to be the authors’ preprint published in bioRχiv. The preprint server name could be given.
  6. The reference No. 52 might need a correction.
  7. Some words may be missing in the last sentence of Conclusions (“… shows promise for use in in HIV-1 LA…”) (page 12, lines 457-458).

Author Response

An excellent paper. Very well designed, executed and presented. The Introduction describes the background and purpose of the study. Methods are thorough. Results and discussion are excellent. Figures are of a high standard. Conclusions are fitting.

Comment 1: Spelling error on Page 6 Line 247 - should be created?

Comment 2: Page 7 Line 291 should be PBMC

Response 1 and 2: We appreciate the positive feedback from Reviewer 1 and thank them for noticing the spelling errors. Page 6 Line 247 “reated” has been changed to “related”. The sentence now reads “The nanometric channel size and related interplay of steric and electrostatic forces acting on molecules diffusing through the nanochannels alters free Fickian diffusion[36].” Further on Page 7 Line 291 it has been corrected to “PBMC”.

Comment 3: One question that I have is that from my experience is that subcutaneous implants are best suited to drugs that only require mcg per day. Eg. contraceptive hormones. Therefore, you can leave Implanon in the body for very long times without intervention. Your implant is at +2mg/day. So what is the maximum life span of your device? Is a short product lifespan a concern?

Response 3: We welcome the reviewer’s comment. We agree with the reviewer that subcutaneous implants are best suited to drugs that only require low doses because implants can have a smaller volume and release for longer periods of time. Consequently, if an implant releases in mg per day, the implant volume needs to increase to be able to load more drug. An alternative is the ability to refill the subcutaneous implant, as we have previously shown in Chua et al. (Journal of Controlled Release 2018). In the current implant design used in this study, the residual drug within the implant ranged 45.31 – 76.34% of the initial loaded amount. With a mean release rate of 3.07 ± 1.10 mg/day, the maximum lifespan is 3 months. Thus, the device design would need to be altered to a refilling device to extend the lifespan and deliver to people living with HIV-1.